# Memory Safe Computations with XLA Compiler

**Artem Artemev**
Imperial College London
Secondmind
a.artemev20@imperial.ac.uk

**Yuze An**
Imperial College London
yuze.an21@imperial.ac.uk

**Tilman Roeder**
Imperial College London
tilman.roeder17@imperial.ac.uk

**Mark van der Wilk**
Imperial College London
m.vdwilk@imperial.ac.uk

## Abstract

Software packages like TensorFlow and PyTorch are designed to support linear algebra operations, and their speed and usability determine their success. However, by prioritising speed, they often neglect memory requirements. As a consequence, the implementations of memory-intensive algorithms that are convenient in terms of software design can often not be run for large problems due to memory over-flows. Memory-efficient solutions require complex programming approaches with significant logic outside the computational framework. This impairs the adoption and use of such algorithms. To address this, we developed an XLA compiler extension[1] that adjusts the computational data-flow representation of an algorithm according to a user-specified memory limit. We show that k-nearest neighbour, sparse Gaussian process regression methods and Transformers can be run on a single device at a much larger scale, where standard implementations would have failed. Our approach leads to better use of hardware resources. We believe that further focus on removing memory constraints at a compiler level will widen the range of machine learning methods that can be developed in the future.

## 1 Introduction

Progress in science is inextricably linked with advances in scientific computing, in terms of both software and hardware. This is particularly noticeable in machine learning through the huge impact of numerical software packages supporting automatic differentiation (Baydin et al., 2018). Packages such as TensorFlow (Abadi et al., 2016), PyTorch (Paszke et al., 2019), or JAX (Bradbury et al., 2018) greatly accelerated **1)** the implementation of gradient-based optimisation procedures by eliminating error-prone manual differentiation, and **2)** the execution of code by leveraging modern and heterogeneous hardware (e.g. GPU, TPU or IPU). A large portion of this impact is attributable to the accessible and user-friendly form that these features were delivered in. This contributed to the growth in the machine learning community, in terms of methodological researchers, as well as the wider scientific audience and practitioners.

The aforementioned software frameworks work by chaining together efficient implementations of mathematical operations (known as *kernels*). By providing implementations that are tailored to various types of hardware, a speed-optimised implementation can be obtained. While speed is certainly important to pursue, many algorithms face a different challenge: hardware memory constraints. Often, these have a larger impact, as memory constraint violations can lead to the execution terminating before an answer is obtained. This make-or-break property is particularly noticeable on GPUs, where

---

[1]The code is available at https://github.com/awav/tensorflow.

allocating more memory than is physically available leads to an immediate termination of execution, and larger amounts of physical memory comes at a significant cost.

Now that numerical computation frameworks are widely used, they strongly influence what machine learning algorithms are adopted. This happens through hard limitations, as well as usability considerations through what is easily implementable. Currently, the emphasis on optimising runtime causes many algorithms to be severely memory limited, or too cumbersome to implement. This is particularly noticeable in methods that rely on linear algebra computations, e.g. kernel methods, nearest neighbour methods for geometric deep learning (Bronstein et al., 2017), or transformers.

In this work, we aim to remove these limitations, by developing a tool that optimises code to be more memory efficient, with a particular focus on linear algebra operations. This optimisation is transparent to the user, and therefore allows many algorithms to be run at scales that were previously impossible, while leaving implementations as simple as before. This allows a wider range of algorithms to take advantage of "the bitter lesson"—*"General methods that leverage computation are ultimately the most effective"* (Sutton, 2019)—while making them more accessible to the wider community, as computational frameworks have sought to do all along.

Our method is implemented as an extension to the XLA compiler (Leary and Wang, 2017), which we chose due to its wide use and support for optimising computations specified in TensorFlow and JAX. We demonstrate the benefits of our method by scaling algorithms where simple implementations do not scale due to memory bottlenecks, such as k-nearest-neighbours, sparse Gaussian process regression (Titsias, 2009), and Transformer (Vaswani et al., 2017). With our extensions, these methods scale to far larger problems, *without changing a single line* of their implementation in Python. Our Gaussian process experiment shows that simply scaling up a 13 year old method can outperform much more recent methods, indicating that older methods may be undervalued in recent literature.

## 2 Motivation: Memory-Constrained Machine Learning

Since memory overflows cause the execution of code to be immediately halted without producing any result, memory constraints form the key obstacle for scaling many machine learning algorithms. Researchers and ML practitioners often reach this limit, as memory is a scarce resource that comes at a considerable cost, particularly in GPUs. This is particularly noticeable in algorithms where minibatching is undesirable, or that rely on pairwise distances, like k-Nearest Neighbours (kNN) or Gaussian processes (Rasmussen and Williams, 2006). Even in modern deep learning, memory constraints cause problems by limiting batch sizes, layer widths, or sizes of attention mechanisms (Vaswani et al., 2017). In all of these examples, linear algebra operations cause the bottleneck. For kNN, kernel methods, and transformers the root of the problem is a pairwise matrix needs to be computed between inputs, giving a quadratic memory cost.

Often, more memory efficient implementations *can* be programmed at the cost of increased software complexity. This ranges from minor annoyances, for example accumulating minibatch gradients in an outer loop for large-batch training, to complex engineering efforts that have been published as scientific contributions in their own right, for example in scaling Gaussian processes to $> 10^5$ datapoints (Gal et al., 2014; Wang et al., 2019a; Meanti et al., 2020).

Our goal is to provide a tool that finds memory-efficient ways to execute algorithms, without the need for increasing software complexity. This will allow scientists and practitioners to access the benefits of scale in existing methods more easily, and without incurring the cost of expensive large-memory hardware. For the main demonstration of our approach, we will automatically obtain a memory-efficient implementation of sparse Gaussian process regression (Titsias, 2009), which was previously implemented with considerable difficulty (Gal et al., 2014). The increase in scale makes the method competitive in comparisons where it was previously dismissed as not scalable enough (Wang et al., 2019a), showing the value of reducing the barriers to scaling.

## 3 Related Work

A popular approach to address memory issues is distributing computation across multiple resources like a group of GPUs or a computer cluster with network protocol connectivity between machines (Buyya, 1999; Dean and Ghemawat, 2008). More specifically, sharding allows large tensors to be

split up and distributed across multiple devices, which increases the total amount of memory available for an algorithm, but comes at the cost of requiring more hardware resources. Most computational frameworks[2] (Abadi et al., 2016; Shazeer et al., 2018; Bradbury et al., 2018; Paszke et al., 2019) support sharding, although some manual specification of how to distribute computation is required. This complicates an implementation and requires the user to have a wide engineering skill set. Automatic sharding tools such as Tofu (Wang et al., 2019b) simplify implementation somewhat, although the specification of computations through a custom interface is still required. While sharding approach does allow scaling of certain implementations, it remains wasteful for algorithms that *can* be implemented in a more memory-efficient way, but where it is simply cumbersome to do so.

Compilers have been introduced to allow humans to express programs in an elegant way, while generating programs that actually run well on specific hardware Aho et al. (2006). Our goal of obtaining memory-efficient implementations, while keeping code convenient for humans, is therefore suited to be addressed by adding memory optimisations to a compiler. Compilers are already being used to optimise computational graphs, notably in JAX, TensorFlow and PyTorch by XLA (Leary and Wang, 2017), TVM (Chen et al., 2018), Glow (Rotem et al., 2018) for PyTorch only. TVM performs similar optimisations to XLA, but unlike XLA, it is not seamlessly integrated into popular frameworks and requires additional user effort.

The optimisations in XLA mainly focus on increasing code speed, for example through *common sub-expression elimination* (CSE), *dead code elimination* (DCE), operations *fusion*, and other more specific modifications. The main advantage of XLA is that it optimises computations in a way that is completely transparent to the user who specifies the computational graph. Although XLA and TVM implement low-level memory optimisations, they do not adapt code handling large tensors to satisfy memory constraints. For the matrix and linear algebra tasks that we consider, KeOps (Feydy et al., 2020; Charlier et al., 2021) currently provides the most efficient memory management. To achieve any benefits, a user must specify a series of computations using KeOps classes, which form a layer above the PyTorch framework. KeOps works similarly to a compiler, by first building a symbolic representation of the computation, which allows the computation to be broken into memory-efficient sections, that are then run with custom CUDA kernels.

In terms of prior work, KeOps is closest in aim and achievement to ours. We aim to address three of its limitations. Firstly, KeOps requires users to reimplement their algorithms using KeOps classes. While the programming interface is elegant, needing to mix KeOps and other computational frameworks does add complexity. Secondly, for KeOps to be able to optimise an operation, it has to be reimplemented within KeOps, which significantly duplicates effort. Finally, because of the former drawback, KeOps does not inherit the support for a wide range of hardware from e.g. JAX/TensorFlow.

## 4  Memory Efficient Matrix and Linear Algebra Operations in XLA

Compilers are a promising way for improving runtime properties of code, without requiring user intervention and while leaving code elegant. The specific matrix and linear algebra optimisations that we consider have not yet been implemented in any of the frameworks discussed above. They *could* be implemented in any of TVM, KeOps, or XLA. We choose to extend XLA over TVM, due to XLA's better integration with common computational frameworks. In addition, we choose to extend XLA over KeOps, because it **1)** does not require algorithms to be rewritten in a separate framework, **2)** can optimise computational graphs in their entirety, rather than just what is implemented in the separate framework, and **3)** can take advantage of the full capabilities that already exist in JAX/TensorFlow.

We introduce several optimisation strategies (known as *optimisation passes* in the XLA codebase) into the XLA optimisation pipeline. We aim to constrain the program's memory footprint with minimal sacrifices in the execution speed. The optimisation passes examine the entire computational data-flow graph (High Level Optimiser Internal Representation, or HLO IR), search for weak spots, and try to eliminate them. Abstractions at a similar level to HLO IR have been shown to be convenient for optimising linear algebra operations (Barthels et al., 2021). We add match-and-replace operations, e.g. to introduce a more efficient distance computation, reshuffling operations for expressions that are invariant to evaluation order, and splitting with large tensors to reduce memory usage.

---

[2]Published under permissive open-source licenses, like Apache or BSD.

**Listing 1** Chain multiplication example $C = ABv$ for $A, B \in \mathbb{R}^{n \times n}$, and $v \in \mathbb{R}^n$.

```
@jax.jit
def matrix_matrix_vector_mul(A, B, v):
    C = A @ B @ v
    return C
```

## 4.1 Match and replace

The *match and replace* optimisation pass, also known as *peephole optimisation* (Aho et al., 2006), searches for expressions in a data-flow graph for which we know in advance that an equivalent and more efficient version exists. For example, we search for expressions that compute Euclidean distance in naive form between vectors of length $n$ and $m$ with a dimension $d$. The naive Euclidean distance computation uses broadcasting over the dimension $d$ and creates a temporary tensor with entries $(x_{nd} - y_{md})^2$ of size $n \times m \times d$. This can be replaced with $\sum_d x_{nd}^2 + y_{md}^2 - 2x_{nd}y_{md}$, where the largest tensor has size $n \times m$.

Replacing sub-parts of the graph is a standard procedure in compilers like XLA, although many linear algebra tricks have been missing. While the Euclidean distance is the only match-and-replace optimisation we implement, other operations can easily be added, for example, efficiently adding diagonals to matrices without allocating a dense square tensor where only the diagonal is non-zero.

## 4.2 Reordering

A computational data-flow graph is an ordered sequence of operations, with the order of operations influencing the memory usage. In some cases, reordering sub-parts of the data-flow graph can lead to reductions in the memory footprint. The classical example of reordering is the optimisation of matrix chain multiplications. For example, consider the matrix expression $C = ABv$ for matrices $A, B \in \mathbb{R}^{n \times n}$, and $v \in \mathbb{R}^n$. In the listing 1, the order of operations determines that the matrix multiplications are performed from left to right, i.e. $C = (AB)v$, which gives the most inefficient execution order with runtime complexity $O(n^3)$ and memory complexity $O(n^2)$. Changing the order to $C = A(Bv)$ improves time complexity to $O(n^2)$ and practical memory complexity because the intermediate multiplication result of $Bv$ is a vector not a matrix as in the case of $AB$ multiplication.

The optimisation of matrix chain multiplication is possible due to the associativity of matrix multiplication, such that the result of the matrix multiplication chain does not depend on where parentheses are placed. There are many efficient and sophisticated algorithms for addressing this task (Chin, 1978; Czumaj, 1996; Barthels et al., 2018; Schwartz and Weiss, 2019). We implement a simple matrix chain multiplication optimisation algorithm using dynamic programming in a bottom-up fashion (Barthels et al., 2018, §2). Our version of matrix chain multiplication algorithm supports expressions with matrix multiplications, sum reductions and transposes.

## 4.3 Data-flow graph splitting

Often, a part of a computational data-flow graph can be divided into multiple independent copies, such that each copy of the data-flow graph or its part act on a slice of the input tensor, and the results are combined afterwards in some fashion. This splitting approach is also known as a MapReduce technique (Dean and Ghemawat, 2008), where a computation is divided into smaller and less expensive parts (map) and then combined into the final result (reduce). The splitting technique is common for distributing the computational load. The focus of existing solutions is on exploiting hardware parallelism or utilising multiple devices. Instead, we use the same techniques for reducing total memory consumption, which is possible because the memory for individual map operations can be freed before the whole result is computed.

An optimisation pass starts by running a depth-first search from the final result of the computation. The operations `dot` or `reduce_*` are special, as they often indicate that a computation involving a large tensor can give a smaller result. Once a `dot` or `reduce_*` operation is marked as fully traversed, we recursively search the traversed paths for operands that are impractically large tensors, until we reach operands that are deemed small enough. Along the way, we keep track of which operations are applied, and along which axes they are trivially parallelisable. The result is a collection of sub-graphs,

**Algorithm 1** High-level description of the depth-first search visitor-handler that splits the data-flow graph up to the reduction `dot` operation. Symbol $\rightsquigarrow$ denotes a directed computational path in the data-flow graph. Steps 9–12 are done recursively traversing back visited operations in the data-flow graph.

---

1: **procedure** HANDLEDOT(dot: HloInstruction)
2:   **if** `output_size(dot)` $\geq$ `tensor_size_threshold`, i.e. dot is splittable **then**
3:     Exit and continue traversing succeeding operations in the data-flow graph and search for size-reducing operation for the output tensor of `dot`.
4:   **if** `dot.rhs` *is not* splittable and `dot.lhs` *is not* splittable **then**
5:     Exit and continue traversing the data-flow graph.
6:   **if** `dot.rhs` $\neq$ `dot.lhs` and both operands *are* splittable **then**
7:     Exit and continue traversing the data-flow graph.
8:   Let `operand_to_split` = `dot.rhs` or `operand_to_split` = `dot.lhs` depending on previous splittability conditions.
9:   Let `split_dims` = $\{d_1, \ldots, d_n\}$, `split_producers` = $\{\text{op}_1, \ldots, \text{op}_m\}$, s.t.
10:     $\circ$ $\forall \text{op} \in$ `split_producers` exists a path op $\rightsquigarrow$ `operand_to_split`
11:     $\circ$ $\forall \text{op} \in$ `split_producers`, `input_size(op)` $\leq$ `tensor_size_threshold`
12:     $\circ$ $\forall \text{op} \in$ `split_producers`, $\exists d \in$ `split_dims` which is splittable on the path op $\rightsquigarrow$ `operand_to_split`
13:   **if** `split_dims` $= \varnothing$ or `split_producers` $= \varnothing$ **then**
14:     Exit and continue traversing the data-flow graph.
15:   Let `best_split_dim` = $d \in$ `split_dims`, and `ops` $\subseteq$ `split_producers`, s.t.
16:     $\circ$ $\min_{d \in \texttt{split\_dims}} \lfloor d \div$ `split_size(operand_to_split, tensor_split_size)` $\rfloor$
17:     $\circ$ $\forall \text{op} \in$ `ops`, the path op $\rightsquigarrow$ `operand_to_split` is splittable on `best_split_dim`
18:   Let `split_size` = $\lfloor$ `best_split_dim` $\div$ `split_size(operand_to_split, tensor_split_size)` $\rfloor$
19:   Create while loop `HloInstruction`, s.t.
20:     $\circ$ The loop iterates splits of size `split_size` at `best_split_dim` of paths `ops` $\rightsquigarrow$ `operand_to_split`
21:     $\circ$ The loop applies `dot` reduction operation on the slice of `operand_to_split`
22:     $\circ$ The slice result of the `dot` reduction operation is put into the replica of the original `dot.result`
23:   Replace `ops` $\rightsquigarrow$ `dot.result` instructions with created while loop.

---

that start at operations that produce large tensors, and end at operations that reduce them again, together with candidate axes that they can be split across. According to some heuristics which ensure appropriate sizes for intermediate results, we then turn this entire computation into a while loop, where each iteration computes a manageable part of the final result (fig. 1).

Checking if the axis is splittable is necessary as not all operations act independently on each dimension. For example, element-wise operations can be split on any axis, whereas the triangular solve operation can be split on "batching" dimensions only. Next, the data-flow graph splitting procedure selects the dimension of the largest size which contributes the most to memory.

As we discussed earlier, the decision about where to split the graph depends on the tensor size. We offer two XLA options to the user for out-of-memory mitigation: *tensor size threshold* and *tensor split size upper bound*. Tensor size threshold is a criterion designed for detecting which operations should be marked as candidates for splitting. Tensor split size upper bound serves as a threshold on the largest allowed chunk size for splitting. These options are set equal by default. The command-line snippet at listing 2 shows how a user would use these options by passing them via an environment variable, and the snippet is indifferent to the machine learning framework used by the script. Minimal user effort is required for using our XLA compiler extension. The user is involved only in defining what the suitable threshold and splitting sizes are.

---

**Listing 2** Example of how a user can set options for the extended XLA using the environment variable.

```
XLA_FLAGS="--xla_tensor_size_threshold=1GB --xla_tensor_split_size=500MB" \
python train.py
```

---

```
I[a] ──▶ │ G │ ──▶  A[..., i:j, ...]  ⤳─▶ │ R │ ──▶  O[...]

I[a] ──▶ │ S │ ──▶  I'[i:j] ──▶ │ G' │ ──▶  A'[..., i:j, ...]  ⤳─▶ │ R' │ ──▶  O'[...]
                                                                              Loop
```

Figure 1: The scheme demonstrates transformation of the data-flow graph on the top to the data-flow graph at the bottom. The graph on the top consists of G and R blocks which are generator and reducer operations respectively, I is the tensor input of G, A is the tensor output of G and O is the tensor output of R. The bracket notation `A[..., a, ...]` means that the tensor A has a dimension of size a and A can have other dimensions. The `i:j` is a slicing operation. A ⤳ R denotes an arbitrary amount of operations in a computational path of the data-flow graph between a tensor A and an operation R. The eXLA splitting optimisation procedure converts the top graph into the loop of independent iterations performing the same chain of operations on a small slice `i:j`.

One strong benefit of our compiler-based solution, is that the computational graph represents the whole pipeline of computations, including forward and backward propagation. Our splitting procedure will be applied automatically, regardless of how many derivatives need to be computed. In addition, our procedure encompasses two splitting schemes that the machine learning literature distinguishes: model-based and data-based splitting schemes of the data-flow graph. Model-based splitting schemes involve partitioning the model over its parameters, whereas data-based splitting schemes batch over inputs. The proposed splitting approach supports both schemes. However, our implementation of model-based splitting supports only splitting memory intensive expressions involving model parameters. It is not able to split model parameters themselves into smaller chunks.

### 4.4   XLA limitations

While we still believe that XLA is the right framework for our extensions, several limitations came to light during implementation. One limitation that is shared with all current frameworks, is that they only have a weak linear algebra type system, where matrices are represented as arrays without additional properties. Solutions that support stronger type systems (Bezanson et al., 2017; Barthels et al., 2021) may be able to implement a wider variety of match-and-replace optimisations.

Another limitation comes from the default memory allocation manager not being aware of memory limits. Its current behaviour is to execute nodes in the computational graph, and therefore allocate any required memory, as soon as the required inputs have been computed. This means that even if tensors are split to manageable sizes, memory overflows can still occur if several are executed simultaneously. To prevent this from happening, we had to use limits that were smaller than our total GPU memory.

## 5   Experiments

This section shows how existing software packages take advantage of our extension to XLA (eXLA). We demonstrate our optimisations on non-parametric k-nearest neighbours (kNN) and sparse Gaussian process regression (SGPR) models, and Transformers.

### 5.1   Matrix-Vector Multiplication

We start by demonstrating the improved efficiency that eXLA offers to large-scale matrix-vector multiplications of the form $y = Kv$, where K is an $n \times n$ kernel matrix, and $y, v \in \mathbb{R}^n$. Such computations are common in Conjugate-Gradients-based Gaussian process approximations (Gibbs and Mackay, 1997; Wang et al., 2019a; Artemev et al., 2021), where $K_{ij} = k(x_i, y_j)$ and $k$ is some kernel function. We choose the common Squared Exponential.

We implement this equation using GPflow (Matthews et al., 2017), a TensorFlow-based package that provides a convenient software interface for Gaussian processes and kernel functions. Without eXLA, the entire K would be stored in memory, leading to a $n^2$ memory cost. This makes running on

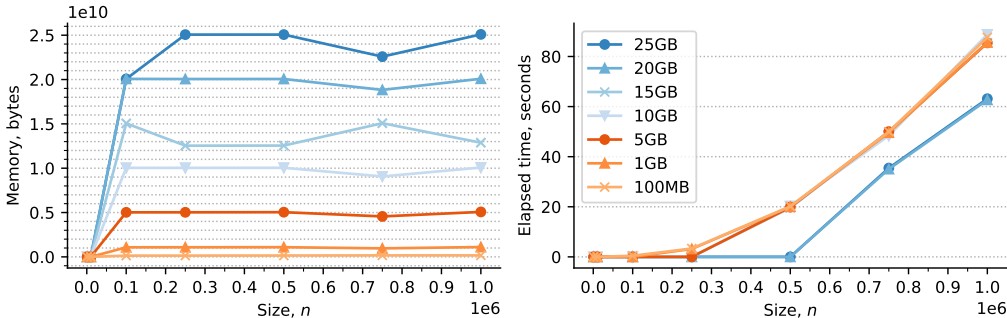

Figure 2: GPU memory consumption and elapsed time of $n \times n$ squared exponential kernel matrix-vector multiplication.

| Dataset | Distance | n | d | KeOps | eJAX | eTF | JAX | TF |
|---|---|---|---|---|---|---|---|---|
| Random | $L^2$ | 1e4 | 100 | 983 263 | 277 364 | 284 777 | 281 695 | 280 826 |
| Random | $L^2$ | 1e4 | 3 | 3 662 188 | 292 804 | 294 971 | 288 098 | 294 776 |
| Random | $L^2$ | 1e6 | 100 | 24 367 | 2433 | 2530 | ∅ | ∅ |
| Random | $L^2$ | 1e6 | 3 | 123 765 | 2512 | 2605 | ∅ | ∅ |
| MNIST | $L^2$ | 6e4 | 784 | 41 084 | 32 290 | 33 455 | 25 544 | 26 138 |
| MNIST | $L^1$ | 6e4 | 784 | 40 697 | 2356 | 2985 | 2498 | 2988 |
| Fashion | $L^2$ | 6e4 | 784 | 40 399 | 32 382 | 33 428 | 25 558 | 26 128 |
| Fashion | $L^1$ | 6e4 | 784 | 40 982 | 2357 | 2984 | 2498 | 2989 |
| Glove-50 | Cosine | 1.18e6 | 50 | 3 464 257 | 2103 | 1929 | ∅ | ∅ |
| Glove-100 | Cosine | 1.18e6 | 100 | 631 420 | 2053 | 1871 | ∅ | ∅ |
| Glove-200 | Cosine | 1.18e6 | 200 | 398 293 | 1967 | 1724 | ∅ | ∅ |

Table 1: Query processing rates (queries per second) for kNN. $n$ and $d$ are the number of data points and the data dimension respectively. Runs which failed due to memory overflow are denoted by ∅. Runs with eXLA are denoted eJAX and eTF respectively.

large datasets infeasible, where e.g. $n = 10^6$ would lead to a memory requirement of 8TB, which is impractical even for the largest of modern GPUs with 40GB of memory. A memory efficient split/implicit implementation is necessary to scale to large datasets, as was impressively done by Wang et al. (2019a), but is cumbersome.

We ran our implementation with eXLA enabled, which allows a user to control the memory of an algorithm. We evaluated the expression in double precision on a Tesla V100 GPU with 32 GB of memory, and applied a range of memory limits. In fig. 2 we report the peak memory consumption and execution time of evaluating the kernel matrix-vector product for different sizes, with different memory limits applied. We see that the memory constraints are not violated, and that dataset sizes are used that are far beyond the 32 GB memory capacity.

## 5.2 K-Nearest Neighbours

K-nearest neighbours is a fundamental machine learning algorithm, with a similar large memory cost. A kNN query selects $k$ closest data points in the dataset to each query point. Brute-force implementations compute pairwise distances between $m$ query points and $n$ data points, resulting in the distance matrix of size $m \times n$. This is followed by a `topk` operation, which is often naively implemented using column-wise `sort` operation on the distance matrix. Our benchmarks show that eXLA scales the brute-force approach and does not fail for large problems, i.e. large $n$ and $m$.

We compare TensorFlow and JAX implementations with and without eXLA optimisations, and a KeOps implementation. We use randomly generated data, common benchmarks like MNIST and Fashion-MNIST, and Glove-50, Glove-100 and Glove-200 from the ANN-benchmark toolkit Aumüller et al. (2020). We use $m = 1e4$ query points in all benchmarks.

| Dataset | Model | RMSE | NLPD | Time (hours) | GPUs |
|---|---|---|---|---|---|
| | SGPR-1000 | $0.0485 \pm 2\mathrm{e}{-}4$ | $-1.603 \pm 4\mathrm{e}{-}3$ | $1.32 \pm 1\mathrm{e}{-}2$ | 1 |
| | SGPR-2000 | $0.0463 \pm 3\mathrm{e}{-}4$ | $-1.649 \pm 7\mathrm{e}{-}3$ | $4.18 \pm 4\mathrm{e}{-}2$ | 1 |
| `houseelectric` | SGPR-3000 | $0.0438 \pm 2\mathrm{e}{-}4$ | $-1.700 \pm 3\mathrm{e}{-}3$ | $8.97 \pm 7\mathrm{e}{-}2$ | 1 |
| | SGPR-4000 | $\mathbf{0.0429} \pm 4\mathrm{e}{-}5$ | $\mathbf{-1.722} \pm 1\mathrm{e}{-}3$ | $15.59 \pm 7\mathrm{e}{-}2$ | 1 |
| | Iterative GP* | $0.054$ | $-0.207 \pm 1\mathrm{e}{-}3$ | $1.55 \pm 2\mathrm{e}{-}2$ | 8 |
| | Iterative GP** | $0.050$ | $\varnothing$ | $79.96$ | 8 |
| | SGPR-1000 | $0.2820 \pm 2\mathrm{e}{-}3$ | $-0.165 \pm 5\mathrm{e}{-}3$ | $0.28 \pm 3\mathrm{e}{-}3$ | 1 |
| | SGPR-5000 | $0.1890 \pm 3\mathrm{e}{-}4$ | $-0.234 \pm 1\mathrm{e}{-}3$ | $5.09 \pm 3\mathrm{e}{-}2$ | 1 |
| `3droad` | SGPR-8000 | $0.1750 \pm 2\mathrm{e}{-}4$ | $-0.303 \pm 1\mathrm{e}{-}3$ | $11.39 \pm 1\mathrm{e}{-}1$ | 1 |
| | SGPR-10000 | $0.1700 \pm 2\mathrm{e}{-}4$ | $\mathbf{-0.321} \pm 4\mathrm{e}{-}4$ | $17.81 \pm 4\mathrm{e}{-}2$ | 1 |
| | Iterative GP* | $0.110 \pm 17\mathrm{e}{-}3$ | $1.239 \pm 25\mathrm{e}{-}3$ | $1.00 \pm 2\mathrm{e}{-}3$ | 8 |
| | Iterative GP** | $\mathbf{0.106}$ | $\varnothing$ | $7.06$ | 8 |

Table 2: The performance of SGPR with inducing points included to the trainable parameter set on `houseelectric` and `3droad` datasets. Iterative GP* and Iterative GP** are trained with lengthscale per dimension and shared lengthscale across dimensions respectively. Iterative GP values are from Wang et al. (2019a), with unreported metrics denoted as $\varnothing$.

Our results are listed in table 3 (see the appendix for a full table that reproduces Feydy et al. (2020, table 3)). In all benchmarks, we set the tensor size threshold for eXLA to 100MB for simplicity, even though this may not be optimal for performance. We observe that eXLA prevents memory overflows in JAX and TensorFlow. In addition, performance is comparable or higher. We acknowledge that KeOps performs significantly better than any JAX or TensorFlow implementation. This is explained by **1)** JAX/TF not having efficient implementations for certain functions (e.g. `topk` runs a full sorting algorithm), and **2)** KeOps having implemented additional optimisations, which could also be added to XLA. However, we note that we also achieved our goal of improving the memory and time performance of a JAX/TensorFlow implementation *without changing the code*.

### 5.3 Sparse Gaussian Process Regression

Gaussian processes (Rasmussen and Williams, 2006) are considered the gold standard method for performing regression with uncertainty estimates. A straightforward implementation requires taking a matrix decomposition of an $n \times n$ kernel matrix (like those considered in section 5.1), which leads to an $O(n^3)$ time cost, and an $O(n^2)$ memory cost. Scaling Gaussian process is challenging, which is often attributed to the time cost. In reality however, large datasets cause memory overflows far before long runtimes become an obstacle.

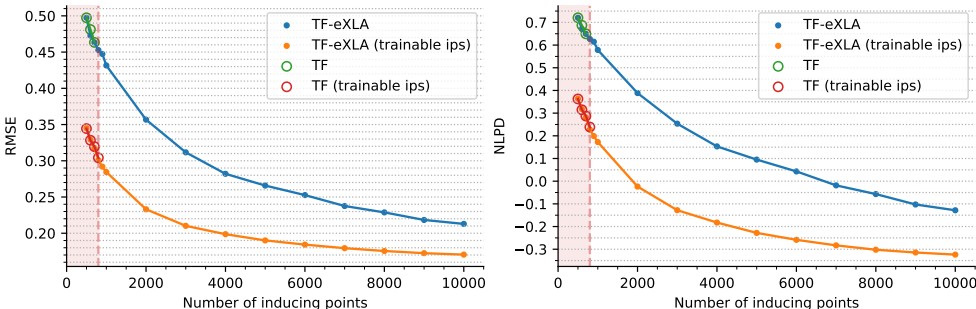

Figure 3: Root mean squared error (RMSE) and negative log predictive density (NLPD) performance test metrics of SGPR for `3droad` dataset as the number of inducing points is increased. The red shaded region emphasizes the capacity of the SGPR model which user can run using standard GPflow and TensorFlow release packages.

Approximate methods have been introduced to deal with both the time and space issues. While there are many variants (Quiñonero-Candela and Rasmussen, 2005), we consider the sparse variational approximation (Titsias, 2009) for which a naive implementation has $O(nm^2 + m^3)$ time cost, and $O(nm + m^2)$ memory cost. Here, $m$ denotes the number of *inducing variables*, which controls the quality of the approximation. Under certain conditions, the method provides reliable hyperparameter selection (Bauer et al., 2016), and very accurate posterior approximations (Burt et al., 2019, 2020) while using $m \ll n$. In practice, these standard implementations may still have their performance limited by how large $m$ can become before a memory overflow occurs. A more memory-efficient implementation with a memory cost of $O(m^2)$ does exist (Gal et al., 2014), but is so cumbersome to implement that it is not widely used or compared against.

Fortunately, the splitting optimisation we implemented in eXLA can discover the same procedure that was engineered by Gal et al. (2014). Moreover, since eXLA operates on the entire computation graph, it optimises gradients as well as the optimisation objective function with no additional effort, while Gal et al. (2014) needed to implement gradients manually. We demonstrate the utility of eXLA by scaling the GPflow (Matthews et al., 2017, 2.3.1 release version) implementation of Sparse Gaussian process regression (SGPR, Titsias, 2009), *without any modifications* of the code.

With our eXLA optimisations, SGPR was able to scale to much larger datasets, with more inducing points. We conduct experiments on a Tesla V100 GPU with 32 GB of memory, and run on two of the largest UCI datasets that are commonly considered in Gaussian process research: `3droad` and `houseelectric` with total number of data points 434,874 and 2,049,280 respectively. We set the tensor size threshold and the tensor split size in eXLA to 1GB. We primarily compare to Wang et al. (2019a), who use a Conjugate Gradients approximation (Gibbs and Mackay, 1997) to achieve the most impressive scaling of a Gaussian process approximation to date, using an impressively engineered implementation that manually splits and distributes parts of the computation.

In fig. 3 we compare GPflow's SGPR implementation with and without eXLA as we increase the number of inducing points. We see that until about 800 inducing points, the normal and eXLA runs result in the same predictive metrics, as desired. After 800 inducing points, runs without XLA fail with an "out of memory" error, while with eXLA we scaled to $10^4$ inducing points. Simply scaling the method in this way leads to significant performance improvements.

We compare predictive accuracies directly with the scalable Conjugate Gradients implementation of Wang et al. (2019a). In that paper, SGPR was discussed as a method that would not scale, probably due to the difficulty of implementing it in a memory-efficient way as in Gal et al. (2014). Table 2 shows that simply scaling SGPR using eXLA makes it competitive with the Conjugate Gradients implementation of Wang et al. (2019a), without needing additional hardware.

## 5.4 Transformers

Neural network architectures based on the attention mechanism, such as Transformers (Vaswani et al., 2017), have become a common tool for solving natural language processing tasks. Transformers have also been applied successfully in computer vision classification tasks (Dosovitskiy et al., 2020) and image generation (Ramesh et al., 2022). The attention mechanism $\text{Att}(Q, K, V) = \text{softmax}\left(\frac{QK^\top}{\sqrt{d}}\right)V$, with $Q, K \in \mathbb{R}^{b \times s \times d_k}$, and $V \in \mathbb{R}^{b \times s \times d_v}$, is an expression that lies in the core of Transformer models. The memory consumption by the intermediate result $QK^\top$ grows quadratically with dimension $s$, i.e. $O(bs^2)$. Rabe and Staats (2021) investigated ways of reducing the memory overhead by sequentially computing parts of the attention operation with large $s$. Their technique is very similar to the eXLA's splitting optimisation pass from section 4.3. The advantage of eXLA is the automatic application of such schemes, depending on sizes of the input dimensions $b$, $s$ and $d$ as well as user's eXLA setting for the tensor size limits.

We test eXLA on a sequence-to-sequence language translation Transformer. We use the configuration of the translation Transformer from the TensorFlow website[3]. We increase the size of the attention mechanism by increasing the size of the randomly generated input sequence until the code fails with a memory overflow; we then show that running the same implementation with eXLA avoids this failure. In the experiment with the language translation Transformer this corresponds to increasing the input sequence. We run experiments on a single Nvidia V100 GPU with 32GB memory. Figure 4

---

[3]`https://www.tensorflow.org/text/tutorials/transformer`

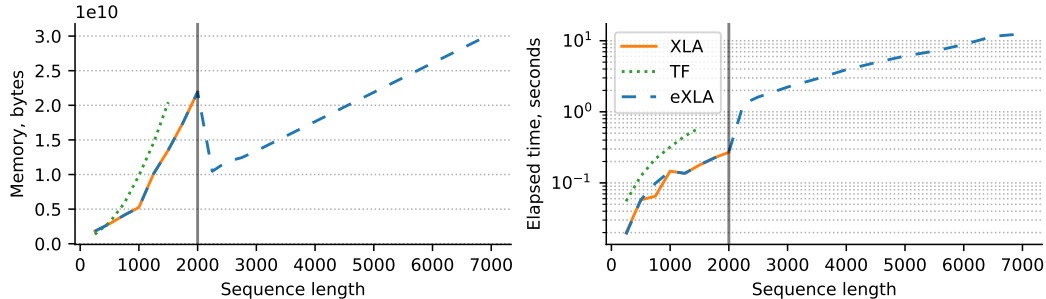

Figure 4: Memory and elapsed time metrics for the language Transformer. The grey vertical line marks the maximum sequence size where implementations in TensorFlow and XLA fail with OOM, but eXLA continues to work.

shows that out of the box implementations of translation Transformer using TensorFlow with its default compiler (TF) and TensorFlow with the XLA backend fail to run when the sequence length larger than 2000. The Transformer model compiled with eXLA optimisations managed to run with sequences up to 7000 with tensor limit set to 10GB and the tensor split size set to 1GB. The drop in the fig. 4 after 2000 is explained by eXLA's splitting optimiser modifying the graph with tensor splits equal to 1GB. After eXLA optimisation applied, there are no tensors in the graph larger than 1GB which can be allocated at the same time together on the GPU. In these situations a practitioner might use larger split size if the hardware allows it, which in turn will boost the computational performance.

## 6 Discussion

We showed that our XLA compiler optimisations (eXLA) prevented memory overflows in algorithms with large tensor or linear algebra operations. Our compiler extension automatically adjusts computational data-flow graphs to control memory utilisation. As demonstrated in the experiments section, machine learning models compiled with eXLA could run at a greater scale, whereas their out-of-the-box implementations failed with Out of Memory errors. Crucially, we used existing software packages without modifying any code.

In addition to showing that our compiler extensions work as intended, our experiments also provide directly useful empirical results for Gaussian processes. We managed to run an "old" method (SGPR, Titsias, 2009), with unchanged code, to obtain empirical results that outperformed a state-of-the art method (Wang et al., 2019a). This shows that existing methods may simply need to be scaled in order to provide high-quality solutions, which are even guaranteed by theory in certain cases (Burt et al., 2020). We also provide proof-of-concept experiments that show that eXLA can help with running Transformers (Vaswani et al., 2017) which are notoriously memory hungry, on cheaper hardware without increasing software complexity.

The exciting possiblity of eXLA is that it opens up the possibility to probe behaviour of machine learning models in regimes that were previously infeasible, and on cheap hardware. For example, one could train very wide neural networks, to empirically compare to behaviour predicted by NTK theory (Lee et al., 2018; Matthews et al., 2018; Jacot et al., 2018; Novak et al., 2020).

The current implementation of eXLA is still only a demonstration of what compiler optimisations could achieve, and many more optimisations can be added. We believe that increasing the capability of compilers like XLA will greatly increase the efficiency of researchers and practitioners. We hope that community-driven compiler projects will contribute to the community in a similar way to how existing numerical frameworks already do.

## Acknowledgements

Thanks to David R. Burt and Sebastian W. Ober for the feedback on the draft of this paper, Fergus Simpson and Alan Saul for their advice on Transformer models. We also would like to thank Lev Walkin and Renat Idrisov for discussions about compilers in the beginning of this project.

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
