# OpenReview forum: "Memory safe computations with XLA compiler"
_NeurIPS.cc/2022/Conference — NeurIPS 2022 Accept_

### Official Review · Reviewer_1TM1 · 2022-07-04

**Rating:** 5
**Confidence:** 3
**Soundness:** 2 fair
**Presentation:** 2 fair
**Contribution:** 2 fair

**Summary:**

To save memory usage in machine learning workloads, this paper paper presents a graph-level rewriting algorithm that is integrated with the XLA compiler. The algorithm is tested against two non-deep learning workloads, k-nearest neighbor (kNN) and sparse Gaussian process regression (SGPR).

**Questions:**

The first confusion point to the reviewer is that the work seems to be focusing very much on deep learning compilation, as mentioned on line 20-21, line 48-49, line 98-100, Section 4.4, all of which are deep learning frameworks and compilers. However, the experiments are only conducted on two much simpler non-deep learning algorithms. Is this possible to demonstrate any experiments with deep learning workloads?

The second question is about literature review. The paper seems to propose that KeOps as major prior work, while ignoring all significant works in memory reduction techniques in deep learning. Is there any consideration not to do so?


**Ethics Review Area:**

["I don’t know"]

**Limitations:**

As mentioned before, the limitation includes three components:
1) The novelty of this work, if only limited to various rewriting, is less significant.
2) While focusing on deep learning compiler and frameworks in paper writing, the experiments are not relevant to deep learning workloads.


**Strengths And Weaknesses:**

Strength:
1) The primary strength of this paper is that it proposes to solve the memory constraint problem, which is an existing painpoint of deep learning systems;
2) The algorithm is integrated with XLA, a mature deep learning compiler, which makes it easier for productionalization;
3) The paper optimizes kNN and SGPR and delivers positive results.

Weakness:
1) There exists quite a lot of work reducing memory footprint in deep learning [1, 2, 3], all of which are quite non-trivial and made significant contributions to the field of deep learning compiler. This paper does not seem to provide comparison with those existing literatures.
2) The algorithms present in this paper, including match-and-replace, reordering and dataflow graph splitting, seems to be very familiar but less novel. For example, changing computation order (Sec 4.2) (AB)v = A(Bv) seems to be a classic trick to save memory.
3) Even though XLA is a deep learning compiler, and also dataflow rewriting (Sec 4.3) is a classic mechanism in deep learning compilation, the experiments are conducted only on much simpler workloads like kNN, which only contains a few matrix operators and does not have a strong motivation to conduct generic graph-level rewriting. Therefore, it's less clear to the reviewer that the proposed algorithm could be a perfect fit for deep learning compilation.

---

> ### Author Response · Authors · 2022-08-02
> **Reply to Reviewer 1TM1 (part 1 out of 2)**
>
> Thank you for your comments and useful feedback.
>
> > There exists quite a lot of work reducing memory footprint in deep learning [1, 2, 3]
>
> We would be interested in seeing your references as it appears you forgot to add the footnotes to your review. Based on our literature review, our solution (eXLA) is unique in the manner in which optimisations are made. eXLA performs its optimisations on the HLO level, and the user can inspect the optimisations made straightforwardly. Also, the memory optimisations are performed automatically given limits on maximum tensor sizes, there is no need for the user to modify their code. Other papers that discuss similar techniques require significant changes in the existing code, and often these techniques do not focus on memory safety.
>
> > The algorithms present in this paper, including match-and-replace, reordering and dataflow graph splitting, seems to be very familiar but less novel. For example, changing computation order (Sec 4.2) (AB)v = A(Bv) seems to be a classic trick to save memory.
>
> We agree that many of the optimisations incorporated in our eXLA optimiser are known approaches for reducing the memory footprint. A key benefit of incorporating these optimisations into the eXLA compiler is that they can then be applied automatically, without the user having to manually tune their code (e.g. by changing software design to incorporate manual optimisations), such as manual ordering of computations. Performing these optimisations manually is inherently error-prone and time-consuming; the eXLA approach alleviates these issues. As such, the user can continue to write their code in a range of mainstream linear-algebra frameworks (e.g., TensorFlow, PyTorch, JAX) and automatically apply our memory optimisations. Our optimisations allow a user to focus more on design choices and think less about efficient computations, as these optimisations are performed automatically.
>
> > Even though XLA is a deep learning compiler, and also dataflow rewriting (Sec 4.3) is a classic mechanism in deep learning compilation, the experiments are conducted only on much simpler workloads like kNN, which only contains a few matrix operators and does not have a strong motivation to conduct generic graph-level rewriting.
>
> XLA is designed to be more general than just for optimising deep learning, which is indicated by its name standing for accelerated linear algebra [1]. Additionally, TensorFlow positions itself as a system for large-scale machine learning [2], and JAX's description emphasises that it is a tool for high-performance machine learning research [3]. kNN and SGPR models are widely used in machine learning.
>
> * kNN plays an important role in geometric problems, 3D computer graphics and it is a sub-algorithm for many existing machine learning models. kNN is used in geometric deep learning [4], for example in models such as Point CNNs [5] and Dynamic Graph CNNs [6].
> * It is arguable whether SGPR is a simple model. SGPR relies on elegant mathematical concepts that allow it to cope with noisy data while indicating how to adjust computational complexity through inducing points. The GPflow implementation is fairly elegant, but the final computational graph is far from simple [8].
>
> ... part 2 in the next comment (references are also in the part 2)

---

> > ### Author Response · Authors · 2022-08-02
> > **Reply to Reviewer 1TM1 (part 2 out of 2)**
> >
> > > The first confusion point to the reviewer is that the work seems to be focusing very much on deep learning compilation, as mentioned on line 20-21, line 48-49, line 98-100, Section 4.4, all of which are deep learning frameworks and compilers.
> >
> > We apologise for not being clear in our initial submission, but our work is focused on improvements to the existing XLA compiler that is widely used by a number of general-purpose machine learning frameworks and linear algebra accelerators, including Tensorflow, PyTorch and JAX. Although these are widely used as a basis for deep-learning frameworks, they are not exclusively applicable to deep-learning, as illustrated in our experiments.
> >
> > > Is this possible to demonstrate any experiments with deep learning workloads?
> >
> > Yes. We tested eXLA on the transformer model implemented in the TensorFlow website tutorial [7]. In our test run we executed Transformer model forward pass on random data. We changed only the input sequence length, which we varied from 250 to 7000. The rest of the configuration of the transformer model is the same as in the tutorial. The experiment was run on one GPU V100 GPU with 32GB. We created a plot [9] running the same code using TensorFlow, XLA, and eXLA compilation configurations. TF and XLA failed to run with sequence lengths of more than >2000, whereas the same code with eXLA was successfully run with sequences >2000.
> >
> > The plot shows the maximum allocated memory on GPU (left) and the elapsed time (right) for Transformer's forward propagation against increasing sequence length. The kink on the left plot at 2000 is where eXLA started applying optimisations. The kink in the left side of the plot at value 2000 is where eXLA started applying optimisations to the graph and smaller memory consumption is a sign that the splitting sizes are too small. The linear memory increase after 2000 indicates that the memory allocator can still allocate multiple tensors at a single point of time during the execution or slow deallocating tensors.
> >
> > * [10] is the Transformer HLO graph before eXLA optimisations.
> > * [11] is the Transformer HLO graph after eXLA optimisations.
> >
> > > The second question is about literature review. The paper seems to propose that KeOps as major prior work, while ignoring all significant works in memory reduction techniques in deep learning. Is there any consideration not to do so?
> >
> > In our related work section, we discuss the frameworks that reduce memory cost and provide some automatic ability to do so. We chose this focus because the contribution of our paper is to automatically perform memory optimisation on a specific method without the need for a user to modify code. It is true that we did not explicitly discuss papers with new deep learning methods that have lower memory costs. These methods require new implementations, and usually introduce approximations which change the output of the code. This lies outside of our contribution, and so we did not discuss them.
> >
> > If there are specific papers that you think we should discuss, please do mention them. We are also willing to consider widening the scope of our related work, if you think this is necessary.
> >
> > [1] https://www.tensorflow.org/xla
> >
> > [2] https://research.google/pubs/pub45381/
> >
> > [3] https://github.com/google/jax#what-is-jax
> >
> > [4] Bronstein, Michael M and Bruna, Joan and LeCun, Yann and Szlam, Arthur and Vandergheynst, Pierre. Geometric deep learning: going beyond Euclidean data, 2017.
> >
> > [5] Li, Yangyan and Bu, Rui and Sun, Mingchao and Wu, Wei and Di, Xinhan and Chen, Baoquan. PointCNN: Convolution on X-transformed points, 2018.
> >
> > [6] Wang, Yue and Sun, Yongbin and Liu, Ziwei and Sarma, Sanjay E and Bronstein, Michael M and Solomon, Justin M. Dynamic graph CNN for learning on point clouds, 2019.
> >
> > [7] https://www.tensorflow.org/text/tutorials/transformer
> >
> > [8] SGPR HLO graph. This is an svg file, please click the Raw button to see a zoomed in version. https://gist.github.com/YToKOq80mh/0349b1d66847f7759a97ee9ee29b7abc
> >
> > [9] Memory and time metrics for Transformer: https://imgur.com/a/R9i7VH8
> >
> > [10] Transformer HLO graph before eXLA. This is an svg file, please click the Raw button to see a zoomed in version. https://gist.github.com/YToKOq80mh/833c0a7ae9b035cd5a2058626e0a73fb.
> >
> > [11] Transformer HLO graph after eXLA. This is an svg file, please click the Raw button to see a zoomed in version. https://gist.github.com/YToKOq80mh/261480d9b116eeefb6dc55878a06d2a5.

---

> > > ### Comment · Reviewer_1TM1 · 2022-08-06
> > > **Thank you for your response**
> > >
> > > Thank you for your detailed explanation! I am convinced by the information the authors provided and am happy to raise the rating from 4 to 5

---

### Official Review · Reviewer_esE8 · 2022-07-09

**Rating:** 6
**Confidence:** 3
**Soundness:** 4 excellent
**Presentation:** 3 good
**Contribution:** 3 good

**Summary:**

Authors propose and implement XLA compiler extension that replaces algorithms using a lot of memory with more memory efficient algorithms. This allows to run computations that use much larger data sizes on a single limited-memory device. Authors demonstrate their tool on kNN and sparse Gaussian process regression methods.

**Questions:**

- It would be good if authors ran their system on a set of models that may or may not contain optimizable code and reported whether the system did not yield any negative results. What I am asking for is a testing run: does the system crash, fail, break the underlying code, slow down the underlying code when run on inputs that it may not expect?
- It would be good if authors explained or walked through Algorithm 1. While it is clear what algorithm is supposed to do, the details as written are hard to follow. Some detailed questions: What exactly does line 5 of the algorithm mean? What does "define best_split_dim" mean in line 13? Line 14 says to iterate over best_split_dim, but best_split_dim was not assigned anything. Line 14 bullets are not clear either.

**Limitations:**

I think that authors adequately addressed the limitations

**Strengths And Weaknesses:**

Strengths:
- Authors implement XLA compiler extension that automatically allows restructuring underlying computation graph to use less memory and therefore execute larger computations on a single device
- Authors demonstrate that the system can run computations that would have failed with out-of-memory without optimization
- Authors demonstrate that the system can be comparable to manual optimization

Weaknesses:
- The optimization is limited and runs much slower than KeOps system (authors acknowledge and explain this limitation)
- Limited explanation of splitting algorithm

---

> ### Author Response · Authors · 2022-08-02
> **Response to Reviewer esE8**
>
> We thank reviewer esE8 for giving valuable feedback. We will address your listed weaknesses first, and then answer your questions.
>
> > The optimization is limited and runs much slower than KeOps system (authors acknowledge and explain this limitation)
>
> Reviewer Mdhs have a similar point.
> The main motivation for this work is users' convenience in overcoming memory bottlenecks. Although KeOps achieves this goal, it is more specific in its application area, such as geometric problems. KeOps focuses on a certain type of operations, like `argkmin`, which is the core of kNN. eXLA, on the contrary as generic as possible. KeOps optimisations would be ineffective for the SGPR model. Also, it is worth mentioning that KeOps is not available for TensorFlow and JAX users, and even PyTorch users would need to modify existing code to benefit from KeOps.
>
> > Limited explanation of splitting algorithm
>
> We will improve the writing of splitting section in the paper to the extent the page limit allows us. We also hope that published code will clarify the algorithm even further. We have updated Algorithm 1, please check the screenshot [2].
>
> > What I am asking for is a testing run: does the system crash, fail, break the underlying code, slow down the underlying code when run on inputs that it may not expect?
>
> So far we have tested eXLA on multiple use cases: sparse Gaussian processes, kNN, and transformers. We also tested MLPs with large number of hidden units (up to 1e6) in the hidden layer with full batch training on CIFAR-10 and CIFAR-100. We tested those models up to the extreme where the out of the box implementation failed with OOM. In a smaller scale we tested the correctness of computation after eXLA optimisations. eXLA results were equal to the results of the original computation compiled only with XLA up to the numerical precision (small pertubations are inevitable when the computational graph changes). eXLA recognises input graphs which are not optimisable and leaves these inputs unchanged. For example the splitting of triangular solve $Lx = y$ is possible only on the batch dimension, i.e. $x, y \in R^{n \times \text{batch}}$. If eXLA's splitting algorithm detects that memory limits set by a user are not satisfied by splitting the batch dimension for the triangular solve, eXLA will not try modify that part of the graph any further. In addition, XLA has safety checks: if eXLA introduced incompatible shapes or float types into the input graph, XLA would report that immediately.
>
> > It would be good if authors explained or walked through Algorithm 1. While it is clear what algorithm is supposed to do, the details as written are hard to follow. Some detailed questions
>
> We will update the Algorithm 1 in the paper with [2]. We hope it adds more clarity.
>
> * Algorithm 1, line 5: When lhs and rhs inputs to the `dot` product are not the same tensor and both lhs and rhs tensors fail to satisfy the user's constraint on memory, the splitting algorithm will not attempt to split the computational path starting from that `dot` product. The `dot` product in that case does not perform size reduction.
> * Algorithm 1, line 13: `best_split_dim` is a dimension over which algorithm performs splitting. The splitting is done on the longest continuous path in the graph that contains that dimension. The splitting path starts at a reduction operation, such as `dot` or `reduce_(sum, max, prod, ...)` and ends at the nodes which satisfy memory limits and generate `best_split_dim` dimension. To define `best_split_dim` we use a heuristic.
> * Algorithm 1, line 14: Once the splitting path and `best_split_dim` in the graph is defined, the algorithm replaces that path with a `for` loop which iterates over slices of the `best_split_dim` dimension. For example, suppose the splitting algorithm decides to split a tensor of shape `[10, 1000, 1]` at the 2nd dimension of size 1000 on stripes of size 100. The 2nd dimension is `best_split_dim`, and a `while` loop will iterate with step size 100 over that dimension.
>     In XLA the `while` loop is a procedure. The nodes at the start of the splitting paths become arguments to the `while` loop procedure. The splitting paths themselves become the body of the `while` loop that act on a slice. The result of the `while` loop is combined according to the reduction scheme.
>
> [1] Chern, Felix and Hechtman, Blake and Davis, Andy and Guo, Ruiqi and Majnemer, David and Kumar, Sanjiv, TPU-KNN: K Nearest Neighbor Search at Peak FLOP/s, 2022
>
> [2] Updated version of the Algorithm 1, https://imgur.com/a/RYEWTxe

---

> > ### Comment · Reviewer_esE8 · 2022-08-05
> > **Thank you authors**
> >
> > Thank you authors for response, explanation, and improved description of Algorithm 1.

---

### Official Review · Reviewer_Mdhs · 2022-07-15

**Rating:** 7
**Confidence:** 4
**Soundness:** 3 good
**Presentation:** 3 good
**Contribution:** 3 good

**Summary:**

This paper extends the XLA compiler with several memory optimizations. The users can run much larger scale computations without changing any code.
The evaluation shows that this exertion enables k-nearest neighbor and sparse Gaussian process regression methods to run at a much larger scale on a single device.

**Questions:**

Tiling/Blocking is a widely used technique to save memory.

This paper [1] applies this idea to self-attention, which is a very important and memory-intensive operator for long sequence transformers. Can the technique in eXLA also automatically discover the tiling pattern in [1]?


[1] Self-attention Does Not Need O(n^2) Memory, arXiv 2021

**Limitations:**

There are other memory optimization techniques such as swapping[2] and gradient-checkpoint/rematerialization[3][4]. These techniques naturally fit into compiler optimizations. You can also try these methods in XLA.

[2] Swapadvisor: Pushing deep learning beyond the GPU memory limit via smart swapping, ASPLOS 2020.
[3] Training deep nets with sublinear memory cost, arXiv 2016.
[4] Dynamic Tensor Rematerialization, ICLR 2021.

**Strengths And Weaknesses:**

Strengths:
- Good direction: integrate memory optimization as compiler passes.
- Impressive results: Run much larger scale computations on a single device without code change.

Weaknesses:
- It is still a prototype with many missing optimizations, as it cannot outperform KeOps

---

> ### Author Response · Authors · 2022-08-02
> **Response to Reviewer Mdhs**
>
> Thank you for your feedback on the paper. We will address your comments point by point, starting with the emphasized weakness.
>
> > ... it cannot outperform KeOps
>
> The focus of our optimisations is memory safety and minimal user involvement. Both KeOps and eXLA satisfy memory safety requirements, but the KeOps is not as generic as eXLA in applying memory optimisations. For example, memory optimisations applied by eXLA to SGPR are not possible with KeOps. The performance of KeOps can be explained by smart code generation using symbolic representations and efficient implementations of reduction schemes like `argkmin` (indices for minimal k elements in an array) followed by the native code compilation. Another drawback of the KeOps is that if users want to benefit from KeOps kNN implementation, users would have to use PyTorch and KeOps interfaces. With eXLA, users have the freedom to select any framework that supports XLA, and they are not constrained to specific interfaces. We will make those points more clear in the paper.
>
> > Tiling/Blocking is a widely used technique to save memory.
>
> Indeed tiling, blocking or splitting are widely used and they are proven to work well. The advantage of our approach is that we can apply these optimisations automatically, without needing the user to change any code. So, while these optimisations are already kown, our solution allows them to be used without additional effort, which we believe will be useful to a large range of people. [1] is a handcrafted solution and not incorporated into a compiler, therefore restricting its usage.
>
> > Can the technique in eXLA also automatically discover the tiling pattern in [1]?
>
> We ran successfully the Transformer model from [5] using eXLA. For the attention block from the transformer model, eXLA generated a similar computation to the proposed computation at [1]. The advantage of our method is that eXLA automatically adjusts the computational graph depending on tensors' sizes and their shapes. I.e., for an attention block with short sequence length and large batch dimension, eXLA could go for splitting the batch dimension. When the sequence length is large, and the batch is small, eXLA could decide to split the computational graph at the sequence dimension.
>
> - [7] is an image of attention HLO IR graph *before* eXLA optimisations
> - [8] is an image of attention HLO IR graph *after* eXLA optimisations
>
> > There are other memory optimization techniques such as swapping[2] and gradient-checkpoint/rematerialization[3][4]. These techniques naturally fit into compiler optimizations. You can also try these methods in XLA.
>
> Thank you for sharing [2, 3, 4]. [2, 3, 4] resemble lazy tensors idea where tensors loaded when needed. We considered that idea in the beginning of the project. However, in practice, implementing lazy tensors in a compiler is very challenging. Lazy tensors would require significant changes in XLA compiler, internals of executor and the memory allocation scheduler in particular. The level on which we introduce eXLA optimisations is too abstract for that feature.
>
> [1] Self-attention Does Not Need O(n^2) Memory, arXiv 2021
>
> [2] Swapadvisor: Pushing deep learning beyond the GPU memory limit via smart swapping, ASPLOS 2020.
>
> [3] Training deep nets with sublinear memory cost, arXiv 2016.
>
> [4] Dynamic Tensor Rematerialization, ICLR 2021.
>
> [5] Tensorflow tutorial on Transformers: https://www.tensorflow.org/text/tutorials/transformer
>
> [6] Chern, Felix and Hechtman, Blake and Davis, Andy and Guo, Ruiqi and Majnemer, David and Kumar, Sanjiv, TPU-KNN: K Nearest Neighbor Search at Peak FLOP/s, 2022
>
> [7] Attention block before eXLA optimisations applied, https://imgur.com/b3hBIHn
>
> [8] Attention block after eXLA optimisations applied, https://imgur.com/grQwYQE

---

> > ### Comment · Reviewer_Mdhs · 2022-08-07
> > **Thanks for your response**
> >
> > It is really cool to see that eXLA can automatically optimize attention. I raised my score to 7

---

### Meta-Review · Area_Chair_1w8b · 2022-08-27

**Recommendation:** Accept
**Confidence:** Less certain

**Metareview:**

This paper implements a set of optimizations on top of the XLA compiler with the explicit purpose of reducing the memory footprint of an algorithm. This is important because while a lot of optimization work in this space has traditionally focused on speed, memory is more commonly the bottleneck when running large computations. The key strengths of the paper are that it presents a genuinely useful artifact and is generally well explained (except for section 4.3 which is not very well explained). The results also show that one can get some significant memory improvements with zero additional effort if your code already compiles with XLA. The main limitations of the paper are limited novelty (none of the transformations are particularly surprising) and limited relevance to NeurIPS---this is really a compilers paper, and it's not even evaluated on any deep learning workloads. These two limitations make this a borderline paper, but all reviewers considered it to be above the bar.

Minor comment: In section 4.1, in the compiler literature, these match-and-replace optimizations are known as peephole optimizations. The term should be at least mentioned.

**Award:**

No

---

### Decision · Program_Chairs · 2022-09-14

Accept